# NGF and Its Role in Immunoendocrine Communication during Metabolic Syndrome

**DOI:** 10.3390/ijms24031957

**Published:** 2023-01-19

**Authors:** Jazmín Samario-Román, Carlos Larqué, Pablo Pánico, Rosa Isela Ortiz-Huidobro, Myrian Velasco, Rene Escalona, Marcia Hiriart

**Affiliations:** 1Neuroscience Division, Cognitive Neuroscience Department, Instituto de Fisiología Celular, Universidad Nacional Autónoma de México UNAM, Mexico City 04510, Mexico; 2Department of Embryology and Genetics, Facultad de Medicina, Universidad Nacional Autónoma de México UNAM, Mexico City 04510, Mexico

**Keywords:** metabolism, metabolic homeostasis, metabolic diseases, neurotrophin, pancreatic beta cell, adipocytes, sex steroid hormones, TrkA, p75^NTR^

## Abstract

Nerve growth factor (NGF) was the first neurotrophin described. This neurotrophin contributes to organogenesis by promoting sensory innervation and angiogenesis in the endocrine and immune systems. Neuronal and non-neuronal cells produce and secrete NGF, and several cell types throughout the body express the high-affinity neurotrophin receptor TrkA and the low-affinity receptor p75NTR. NGF is essential for glucose-stimulated insulin secretion and the complete development of pancreatic islets. Plus, this factor is involved in regulating lipolysis and thermogenesis in adipose tissue. Immune cells produce and respond to NGF, modulating their inflammatory phenotype and the secretion of cytokines, contributing to insulin resistance and metabolic homeostasis. This neurotrophin regulates the synthesis of gonadal steroid hormones, which ultimately participate in the metabolic homeostasis of other tissues. Therefore, we propose that this neurotrophin’s imbalance in concentrations and signaling during metabolic syndrome contribute to its pathophysiology. In the present work, we describe the multiple roles of NGF in immunoendocrine organs that are important in metabolic homeostasis and related to the pathophysiology of metabolic syndrome.

## 1. Introduction

Metabolic syndrome (MS) comprises a group of metabolic alterations that increase the risk of developing cardiovascular diseases (CVD), type 2 diabetes mellitus (T2DM), neurodegenerative diseases, and several types of cancer [1,2,3]. According to the WHO, the diagnosis of MS includes the presence of three or more of these signs: central obesity, dyslipidemia (low levels of high-density lipoproteins (HDL), or hypertriglyceridemia), arterial hypertension, impaired fasting glucose levels, and insulin resistance [1,4,5]. Multiple genetic and environmental factors contribute to the development of MS [6].

During MS development, the hypertrophic adipose tissue increases the release of free fatty acids (FFA), which may contribute to the development of peripheral insulin resistance and impaired insulin secretion by pancreatic beta cells [1,5]. In addition, adipocytes increase adipokine secretion, including nerve growth factor (NGF), interleukin 6 (IL-6), IL-1, tumor necrosis factor- α (TNF-α), and monocyte chemoattractant protein-1 (MCP-1) [7,8,9]. The macrophages settled in adipose tissue promote the secretion of proinflammatory cytokines, inducing the development of a low-grade systemic inflammation, which also contributes to inducing insulin resistance and other cardiometabolic alterations [5,10]. In addition, the development of MS includes chronic hyperactivity of the sympathetic nervous system, dysfunction of the hypothalamus–pituitary–adrenal (HPA) axis, and impaired levels of NGF that depend on the stage of the condition [11,12,13]. Specifically, at initial stages of MS (when only three signs are present), there are high levels of circulating NGF, while after developing more than four signs at the same time, the circulating levels of NGF decrease (Table 1) [13,14]. Thus, it is proposed that there are pathophysiological relationships between NGF levels and the development of MS.

NGF is a pleiotropic growth factor secreted by different cell lineages of the neuroimmunoendocrine system and is involved in whole-body metabolic homeostasis [15,16,17,18]. NGF regulates several pathways that could contribute to MS. Nevertheless, an integrative view of the multiple roles of this neurotrophin on tissues involved in the immunoendocrine axis, and consideration of whether their deregulation could contribute to the development of MS, are lacking. To remedy that, we reviewed the literature for the actions of NGF signaling in pancreatic beta cells, adipose tissue, immune system components, and sex hormone secretion, as well as its potential implications during MS development.

**Table 1 ijms-24-01957-t001:** Metabolic syndrome is associated with alterations in NGF levels.

Condition	NGF	Reference
Control F	Plasma levels of NGF vary throughout the hormonal cycle Early follicular phase—low levels Mid luteal phase—high levels	Martocchia et al., 2002 [19]
MS (F and M)	Plasma levels of NGF decrease in MS	Chaldakov, Fiore, et al., 2001 [20]
MS (F and M)	Plasma levels of NGF decrease in MS	Chaldakov et al., 2004 [12]
MS (F and M)	Plasma levels of NGF increase in early stages of MS Plasma levels of NGF decrease in late stage of MS	M. Hristova and Aloe, 2006 [13]
MS (F)	Plasma levels of NGF and subcutaneous adipose tissue expression of NGF increase in MS	Atanassova et al., 2014 [11]
MS (F)	Plasma levels of NGF increase in early stages of MS Plasma levels of NGF decrease in late stage of MS In MS patients with metformin treatment, plasma levels of NGF decrease In MS patients with metformin/aspirin/Diclac treatment, plasma levels of NGF increase	M. G. Hristova, 2011 [14]
MS (F)	Plasma levels of NGF increase in MS patients with overweight, obesity, or morbid obesity	Bulló et al., 2007 [21]
T2DM	Plasma levels of NGF decrease in T2DM Plasma levels of NGF decrease in T2DM + diabetic neuropathy	Sun et al., 2018 [22]
Obesity (F) (BMI > 30; hyperinsulinemia)	Plasma levels of NGF increase in obesity	Molnár, 2020 [23]

MS: metabolic syndrome; F: female; M: male; T2DM: type 2 diabetes mellitus; BMI: body mass index.

## 2. NGF Signaling

The precursor of mature NGF (proNGF, a molecule of 32 kDa) is synthesized as a homodimer, which is cleaved by extracellular proteases (such as plasmin and neuroserpin) or by proteases residing in the Golgi apparatus and secretory vesicles (such as furin and pro-convertase), resulting in mature NGF (13.2 kDa) [24,25]. The maturation process and the ratio of secreted proNGF/NGF depend on the cell type and the physiological context, and the impairment of proNGF relates to the development of several diseases [26,27,28]. Interestingly, there are different NGF species with different molecular weights (40 kDa, 53 kDa, 60 kDa, and 73 kDa), but their specific nature and function are unknown [29,30,31]. Two possibilities are the existence of proNGF variants or diverse mechanisms for posttranslational processing from a unique form of proNGF [29].

Two different receptors bind ProNGF and NGF, namely TrkA (a receptor with tyrosine kinase activity) and p75 neurotrophin receptor (p75NTR, a member of the tumor necrosis factor receptor (TNFR) superfamily). These receptors differ in their signaling pathways and cell effects (Figure 1) [15,16,17,18,19,20,21,22,23,24,25,26,27,28,29,30,31,32]. NGF exhibits high-affinity binding and selectivity to TrkA and binds with lower affinity to the p75NTR. On the contrary, proNGF shows a higher binding affinity for p75NTR [33].

The binding of NGF to TrkA induces its dimerization and transphosphorylation at tyrosine residues located in the TrkA cytoplasmic domains [34]. For instance, phosphorylation at Y785 induces the activity of its effector protein phospholipase C-gamma (PLC-γ), promoting cell differentiation and survival [34,35]. On the other hand, phosphorylation at Y490 activates the MAPK and phosphatidylinositol 3-kinase (PI3K)/AKT pathways [32,35,36], inducing cell proliferation, survival, and growth [37,38].

In contrast, the binding of NGF to p75NTR induces conformational changes in the intracellular domain of the receptor, which recruits effector proteins, including TNF-receptor-associated factor 6 (TRAF6) and receptor-interacting-serine/threonine-protein kinase 2 (RIPK2 or RIP2) [39,40]. Similar to the activity of TrkA, TRAF6 and RIPK2 promote cell survival by activating NF-κB and PI3K/Akt pathways. p75NTR may promote TrkA activity since the association of both receptors allows the formation of high-affinity binding sites for NGF [41]. p75NTR forms heterodimers with sortilin, which induces pro-apoptotic pathways through c-Jun N-terminal kinase (JNK) and activation of caspases 3, 6, and 9 [42,43,44]. Furthermore, the interaction between p75NTR and sortilin increases the affinity of p75NTR to proNGF, thus implying that exposure to high levels of proNGF could lead to pro-apoptotic pathways. In addition to these classical pathways related to proliferation, survival, and apoptosis, NGF receptors participate in metabolic homeostasis [45,46]. Therefore, the proNGF/NGF ratio and alterations in TrkA and p75NTR abundances in peripheral tissues could lead to metabolic abnormalities, as detailed in the following sections.

## 3. NGF in Pancreatic Beta Cells

Pancreatic beta cells are the only cells in mammals that synthesize and secrete insulin. Beta cells also secrete NGF, amylin, gamma-aminobutyric acid (GABA), and other growth factors [17]. Glucose is the main secretagogue for insulin. However, beta cells also express autoreceptors to their chemical messengers such as NGF, which contributes to insulin secretion regulation. Interestingly, both NGF and insulin are secreted in response to high glucose plasma levels [17,18], suggesting an autocrine feedback loop in insulin secretion through NGF.

Pancreatic islets are composed of endocrine, vascular, and immune cells and neurons from the peripheral nervous system (PNS), all immersed in an extracellular matrix. Interestingly, most of these cell types also secrete NGF (Table 2) [47], and their complex relationship determines islet functions [48]. Experimental evidence suggests that NGF contributes to islet structure, maturation, and function.

Pancreatic NGF signaling is essential for islet morphogenesis, innervation, and vascularization [49]. In mice, NGF is robustly expressed in pancreatic vascular cells, and its secretion depends on the glucose concentration. Vascular-specific deletion of NGF and pancreatic-specific TrkA deletion impair glucose tolerance and attenuate glucose-stimulated insulin secretion (GSIS) [50]. In addition, autocrine and paracrine NGF action mediated by TrkA may induce local secretion of epidermal growth factor (EGF) and vascular endothelial growth factor (VEGF), promoting islet survival [49,50]. Immature human and rodent beta cells display lower GSIS compared to adult islets [51], and incubation with NGF increases GSIS in immature beta cells from young rats [52] (Figure 2).

As mentioned above, NGF signaling enhances GSIS in mature beta cells, and pharmacologic inhibition of TrkA prevents the effect of NGF on insulin secretion [49,53]. Activation of TrkA enhances exocytosis of insulin granules, probably through the reorganization of actin filaments by Rac1 activation (Figure 2) [50]. Some of the mechanisms involved in the NGF insulinotropic effect include the upregulation of sodium and calcium channels and increased currents of L-type calcium channels [53,54,55]. Notably, these effects depend on the duration of the stimulation with NGF.

**Table 2 ijms-24-01957-t002:** NGF and its receptors are expressed in cells of the immunoendocrine axis implicated in the development of metabolic syndrome.

Cell Type	NGF	Receptor	Effects Possibly Related to the Development of MS
Pancreatic vascular cells	NGF expression and secretion [50]		NGF deletion impairs glucose tolerance and attenuates GSIS [50]
Pancreatic beta cells	NGF expression and secretion [17,56]	TrkA and p75^NTR^ [49,53]	Low NGF levels induce apoptosis [47,57]
WAT	NGF expression and secretion [8,58,59]		Stimulates lipolysis Induces browning Controls adipokine production Induces inflammatory response Upregulates PPARγ expression Regulates LDL receptor signaling [45,60]
BAT	NGF expression and secretion NGF [60,61,62]		Induces thermogenesis [7,31]
Adipocytes	NGF expression and secretion NGF	p75^NTR^ [8,58,59]	Impair GLUT4 translocation Decrease lipolysis Induce resistance to catecholamines [45,46]
Mast cells	NGF expression and secretion [63] of proteases involved in NGF maturation [64]	TrkA and p75^NTR^ [64]	Mast cells residing in atherosclerotic lesions express high levels of P75^NTR^ and low levels of NGF [65]
Monocytes		TrkA [66]	Modulate the secretion of LPS-induced cytokines [66,67]
Macrophages	NGF expression and secretion NGF [68]	TrkA and P75^NTR^ [68,69]	Resident in adipose tissue and modulate proinflammatory cytokines [70]
Eosinophils	NGF expression and secretion NGF [71]	TrkA [72,73]	During cold exposure, NGF secreted by WAT-residing eosinophils induces intra-adipose axon growth and adipocyte browning [74]
Somatic/interstitial cells (ovary)		TrkA and P75^NTR^ [75,76]	NGF induces follicular growth and androgen secretion by theca cells [76,77]
Granulosa cells (ovary)	NGF expression and secretion, and it accumulates in the follicular fluid [75,76]		Hyperandrogenism has been linked to insulin resistance in the pathogenesis of PCOS [78,79]
Somatic and meiotic germ cells (testis)	NGF expression and secretion [80,81,82]	P75^NTR^ [80,81,82]	NGF induces steroid hormone synthesis [80,81,82] Low androgen production has been related to increased risk of developing T2DM [82]

The discovery that rat [17] and human beta cells [56] synthesize and secrete NGF, along with the notion that NGF regulates GSIS, fueled ideas about the possible roles of this factor during the pathogenesis of MS and diabetes mellitus (DM).

In T2DM, impaired beta cell insulin secretion and sensitivity might be induced through several mechanisms, including endoplasmic reticulum (ER) stress, oxidative stress, inflammation, cytokine secretion, and hyperglycemia, causing beta cell dysfunction and apoptosis [83,84,85]. In vivo and in vitro models show a correlation between low NGF levels and pancreatic beta cell apoptosis [57,86]. Paradoxically, db/db mice show increased plasma NGF levels during the early stages of diabetes [87]. This suggests that NGF levels increase at early stages and decrease at later stages of diabetes and that they could relate to beta cell functions.

Similarly, beta cell depletion by streptozotocin (STZ) treatment induces an initial short-term increase in NGF mRNA expression and secretion by beta cells both in vivo and in vitro [88]. In contrast, after STZ treatment, the long-term levels of pancreatic NGF are reduced in rats. STZ induced a slight downregulation of TrkA and p75^NTR^ [89]. Along with the epidemiological data shown in Table 1, the available evidence points out that pancreatic NGF secretion depends on the stage of the metabolic disease and the degree of beta cell damage.

One possible explanation for the increase in NGF secretion during beta cell damage is related to the antiapoptotic and insulinotropic effects of NGF, acting as a protective factor for beta cells during metabolic stress. Consistent with this idea, pretreatment with 4-methyl catechol (4-MC, an inducer of NGF synthesis) before STZ prevented the chronic decrease in pancreatic NGF levels and reduced beta cell apoptosis. Mechanistically, these effects could be related to the increased NGF synthesis by beta cells and by downregulating the Ras-pathway effectors RASSF1 (Ras association domain family) and NORE1 (member of the Ras association domain family) [47]. Likewise, NGF withdrawal induces apoptosis of beta cells by reducing the activity of survival pathways, such as EGF and insulin secretion, the PI3K/AKT pathway, imbalance in the proapoptotic factor Bad and the antiapoptotic factor Bcl-XL, and activation of JNK [47,86]. Therefore, NGF effectively prevents beta cell loss and induces their survival.

Altogether, the available evidence highlights the relevance of NGF signaling for the islet physiology in physiological and pathological conditions, both contributing to GSIS and beta cell survival. Thus, NGF could be used as a therapeutic agent to treat T2DM [90].

## 4. NGF in Adipose Tissue

Adipose tissue is highly heterogeneous, composed of mature adipocytes and stromal vascular cells, including preadipocytes, fibroblasts, endothelial cells, and immune cells [91]. According to differences in the morphological and metabolic characteristics, adipose tissue is classified as white adipose tissue (WAT), brown adipose tissue (BAT), or beige adipose tissue [92]. WAT is characterized by the presence of adipocytes with a single lipid droplet, serving as an energy store that releases lipids into the circulation when the energy demand increases and the energy substrate availability decreases. In contrast, BAT comprises mitochondria-rich adipocytes with several tiny lipid droplets. This kind of adipose tissue shows a higher vascularization than WAT. BAT’s primary function is to release energy as heat in a process termed thermogenesis. Beige adipocytes share BAT characteristics but are immersed in WAT depots. This led to the “browning” hypothesis, which suggests that WAT adipocytes transdifferentiate from BAT-like adipocytes after exposure to cold or exercise [62,93].

Adipose tissue is an endocrine organ that releases lipids, mainly free fatty acids, prostaglandins, and sterols, and also produces and secretes a wide variety of peptides termed adipokines. Recent proteomic and transcriptomic analyses revealed that more than 600 adipokines could be secreted by WAT [91,94]. These include cytokines, enzymes, growth factors, hormones, and neurotrophins such as NGF (Table 2) [8,58,95]. Noteworthy, adipose tissue synthesizes, secretes, and responds to neuropeptides and neurotrophic factors (Table 2), and it is proposed as part of the diffuse neuroendocrine system [58,62].

The origin of NGF in adipose tissue is still controversial. Some authors suggest that mast cells, immune cells that synthesize NGF, are the primary source of NGF in adipose tissue [96]. However, NGF mRNA is detected by RT-PCR in both mature adipocytes and stromal vascular cells of mice and human WAT [59]. Before and after differentiation into adipocytes, in vitro studies in 3T3-L1 cells and human adipocytes show they express and secrete NGF. Noteworthy, NGF secretion is higher in 3T3-L1 cells before differentiation [7,59], indicating that adipocytes are an active source of NGF. There are differences in NGF mRNA levels between WAT depots. These differences in NGF expression may be related to the proportion of mature adipocytes and stromal vascular cells, the degree and activity of sympathetic innervation, and the local inflammatory status observed in each fat depot [59]. On the other hand, there is sufficient evidence supporting the notion that NGF is synthesized and secreted by BAT [60,61,62].

NGF actions in adipocytes are mediated by TrkA and p75NTR receptors [58]. While an Expression Atlas Database analysis indicates that TrkA mRNA is not expressed in human adipose tissue [58], studies in murine WAT have shown that p75NTR and TrkA receptors are expressed in mature adipocytes and the stromal vascular fraction, highlighting that TrkA mRNA appears relatively weak in mature adipocytes [59]. This has led to the hypothesis that TrkA is transferred to adipocytes from neuronal exosomes [58,97], but the mechanism is unclear. On the other hand, BAT from rats and humans expresses p75NTR and TrkA receptors [61,98]. Future studies are needed to clarify whether these receptors are expressed in WAT adipocytes.

The effects of NGF signaling on adipose tissue are the results of many processes, including those maintaining the adipose tissue architecture, development, and metabolism. Given the central role of adipose tissue, NGF contributes to systemic metabolic homeostasis [45,46]. NGF mRNA and protein expression increase during the first 20 postnatal days in rat peripancreatic and gonadal WAT. During this developmental window, NGF participates in the angiogenesis and vascularization of WAT [95]. Moreover, NGF regulates the development and maintenance of sympathetic innervation of adipose tissue [7,58,62]. Specifically, sympathetic activity stimulates lipolysis in WAT, induces thermogenesis in BAT, and favors the cold-induced browning process of WAT [31]. Furthermore, NGF controls the adipocyte number, adipokine production, and adipose tissue inflammatory response [7] (Figure 3).

The metabotropic effects of NGF on adipose tissue include upregulation of the peroxisome proliferator-activated receptor gamma (PPARΥ), which stimulates lipid uptake and adipogenesis. NGF also regulates the expression of low-density lipoprotein (LDL) receptor-related proteins [60]. Plus, NGF prevents the downregulation of caveolin-1 induced by high glucose, contributing to improved neurotrophin receptor and LDL receptor signaling [60]. In addition, NGF regulates the synthesis of leptin and adiponectin in WAT, driving anorexigenic signals in the central nervous system [8].

Lobules are structural units within the organization of adipose tissue composed of two extracellular matrix compartments, named the septum and stroma. These compartments delimit niches of precursor cells characterized by the expression of p75NTR. Septa are mainly composed of a loose fibrillar network of collagen and elastin, along with progenitor cells CD45−/CD34+/CD31−, which exhibit high expression of p75NTR [99]. Upregulation of p75NTR and NGF might participate in differentiating CD34+ progenitor cells into myofibroblasts induced by TGFβ in subcutaneous adipose tissue. Conversely, TGF1β contributes to p75NTR activation by inducing the proteolytic cleavage of the receptor by the gamma-secretase complex, releasing its intracellular domain. Myofibroblast differentiation in adipose tissue contributes to tissue fibrosis and affects its expandability, thus determining tissues’ growth by hyperplasia or hypertrophy during obesity. Concordantly, hypertrophic adipocytes and adipose tissue fibrosis further increase adipose tissue damage during obesity [99].

A high-fat dietary intake induces p75^NTR^ overexpression in WAT from mice, indicating that obesity can modulate NGF signaling [46]. One possible consequence of dysregulated p75^NTR^ in adipocytes could be direct binding of this receptor to Rab5 and Rab31 GTPases, altering GLUT4 trafficking induced by insulin [45]. Accordingly, the high levels of p75^NTR^ in WAT from obese subjects can contribute directly to impaired GLUT4 translocation to the membrane. Similarly, p75^NTR^ binds to PKA and suppresses its activity, decreasing lipolysis, lipid oxidation, and thermogenesis, and induces resistance to catecholamine signaling [46]. Therefore, p75^NTR^ induced by obesity can directly interfere with insulin-stimulated glucose uptake, lipid metabolism, energy expenditure, and thermogenesis in adipose tissue.

In obese individuals, hypertrophic WAT promotes a proinflammatory state characterized by altered secretion of adipokines. Several studies have shown that NGF constitutes one of the proinflammatory cytokines produced by obese adipose tissue. Furthermore, proinflammatory factors secreted by adipose tissue, such as TNF-α, lipopolysaccharides, prostaglandins D2 (PGD2), and prostaglandin J2 (PGJ2), induce NGF production in adipocytes [8,21,96]. There is a positive correlation between the synthesis of TNF-α with NGF mRNA expression in epididymal WAT and BAT from obese ob/ob mice [59]. In fact, women with metabolic syndrome show high levels of NGF in plasma and subcutaneous adipose tissue [11].

Similarly, NGF levels in murine WAT and BAT increase during stress and diabetes [92]. These works suggest that NGF is directly involved in adipose tissue dysfunction or is a consequence of inflammation in obese adipose tissue. Nevertheless, circulating NGF levels are reduced during the advanced stages of MS. These discrepancies are likely due to the different stages of progression of the disease (initial stages of metabolic syndrome vs. later phases), interaction with other organs, and the presence of other risk factors for metabolic syndrome, which deserves further research. Plus, it is necessary to clarify the specific contribution and mechanisms of NGF during WAT inflammation in future studies.

On the contrary, anti-inflammatory factors, including dexamethasone, rosiglitazone (an agonist of PPARγ), and IL-6, inhibit NGF synthesis by fat cells [7,60]. The endogenous ligand of the PPARγ 15-deoxy-Δ12,14-PGJ2 (15d-PGJ2, a metabolite derived from PGJ2) reduces NGF synthesis and secretion in adipocytes. It is an anti-inflammatory factor that suppresses pathways related to adipose tissue inflammation associated with obesity. Researchers have proposed that the effects of 15d-PGJ2 on the synthesis and secretion of NGF by adipose tissue could be due to the selective expression of the PPARγ2 receptor [7].

## 5. Immunomodulatory Effects of NGF and Its Metabolic Implications

The immune system orchestrates the response to infectious agents and damaged body components, thus influencing the development and homeostasis of the organism [100,101]. Immune system components have bidirectional interactions with the neuroendocrine axis by releasing cytokines, neuropeptides, and growth factors and expressing multiple receptors to hormones, adipokines, neurotransmitters, and neuropeptides, including NGF and its receptors (Table 2) [100]. Notably, deregulation of the immune system is an essential factor in the pathophysiology of MS. As discussed in the previous section, obesity induces low-grade chronic inflammation, which contributes to developing insulin resistance, cardiovascular alterations, dyslipidemia, liver steatosis, and hypertension [102]. On the other hand, MS patients have compromised immunosurveillance, increasing their risk of developing certain forms of cancer and infection [103,104,105]. Although most cells of the immune system secrete and respond to NGF, an exhaustive review of the effects of this neurotrophin is beyond the scope of this work. Instead, we review some of the effects of NGF on individual immune system components that could participate directly in the development of MS.

Mast cells reside in epithelial, mucosal, nervous, and connective tissues, which regulate angiogenesis and local immune system homeostasis [106]. Mast cells were the first non-neuronal cells shown to express, store, and secrete biologically active NGF of high molecular weight (73 kDa) [31,107]. Nevertheless, the nature and function of these high-molecular-weight isoforms of NGF are currently unknown. Mast cells also produce and secrete the protease tryptase, which cleaves proNGF and mature NGF, altering the synthesis and degradation of this neurotrophin [64]. Since mast cells are widely distributed in tissues essential for metabolic and cardiovascular homeostasis, they could modulate the alterations in NGF levels observed in these tissues.

In MS, atherosclerotic lesions have increased expression of p75NTR and mast cell abundance, while NGF levels are decreased [65]. This could be related to the chemoattractant effects of NGF on mast cells, which can exacerbate the inflammation of atherosclerotic lesions. Additionally, mast cells participate in adipose tissue hypertrophy, inflammation, and insulin resistance [106,107]. Nevertheless, it has not yet been described whether NGF regulates adipose tissue-residing mast cells in obesity. So far, it is understood that NGF signaling is important for mast cell differentiation and chemoattraction and potentiates degranulation induced by IgE and substance P (Figure 4) [18,108,109,110,111].

Monocytes are precursors of macrophages and dendritic cells [112]. Stimulation of monocytes with NGF suppresses the synthesis of proinflammatory cytokines induced by lipopolysaccharide (LPS) while it induces the secretion of anti-inflammatory cytokines such as IL-10 and IL-1ra [66] (Figure 4). This anti-inflammatory effect of NGF could be particularly relevant during the development of chronic low-grade inflammation during obesity and MS. Stimulation with LPS induces TrkA expression, and Toll-like receptor signaling can crosstalk with the TrkA pathway [66]. This neurotropic factor also reduces the antigen-presenting capacity of monocytes [113].

Eosinophils participate in the defense against parasites, allergic and asthmatic reactions, and regulating the adaptive immune response, tissue repair, and neural activity [114]. Eosinophils produce NGF in the basal state, and its production is enhanced after co-stimulation with IL-5 and soluble IgA [71]. Conversely, TrkA is necessary for eosinophil and neutrophil recruitment to inflamed tissues [72,73]. Along with the effects of NGF on monocytes and macrophages, these studies highlight the ambivalent nature of NGF during inflammatory processes, which depends on the specific context and the presence of other signals. Potentially, this could explain the discrepancies of NGF levels between early and late MS.

Of note, NGF secreted by eosinophils participates in WAT metabolism and beiging. Beige adipocytes are transdifferentiated from WAT after cold exposure, adrenergic signaling, and exercise. Beige adipocytes express uncoupling protein-1 (UCP-1), have an increased number of mitochondria, and have high metabolic rates [70]. Cold exposure increases NGF secretion by eosinophils residing within inguinal WAT [74]. Plus, increased WAT NGF levels induce intra-adipose axonal outgrowth and WAT browning [74]. Sympathetic stimulation of stromal cells indirectly promotes eosinophil NGF production [74], showing the bidirectional interaction between neuroendocrine and immune systems in the control of metabolism in response to environmental cues. Remarkably, eosinophil-specific deletion of NGF prevents sympathetic innervation and WAT browning, demonstrating the relevance of this mechanism [74]. These findings suggest that NGF agonists could be used to induce adipocyte browning, improving the metabolic profile of obese adipose tissue in obese patients. Nevertheless, therapeutic approaches must consider the differences between systemic and local effects of NGF to avoid unwanted consequences such as macrophage and monocyte activation when there is metabolic endotoxemia.

## 6. NGF in the Secretion of Sex Steroid Hormones

The hypothalamus–pituitary–gonad axis receives several environmental, metabolic, and neuroendocrine inputs, and the synthesis and secretion of steroid hormones are modulated by neurotrophins such as NGF. Furthermore, sex hormone signaling regulates diverse metabolic processes, and alterations related to MS have sex dimorphism. Hence, some metabolic actions of NGF might be mediated by controlling steroid hormone secretion in health and disease.

Both NGF receptors, p75NTR and TrkA, are expressed in human (fetal and adult) [75] and rodent (juvenile and adult) ovaries [76]. These are receptors located primarily in oocytes and granulosa cells from growing follicles (i.e., primordial, primary, and secondary follicles), suggesting that NGF participates in follicular growth and recruitment. Indeed, NGF-/- knockout mice show fewer growing follicles (primary and secondary). These mice display normal concentrations of follicle-stimulating hormone (FSH) and luteinizing hormone (LH), indicating that the observed growth arrest is not due to gonadotropin deficiency but instead due to diminished somatic cell proliferation [76].

The most abundant source of follicular NFG is located in granulosa cells. Granulosa cells from antral follicles produce proNGF and secrete it into the follicular fluid, which is cleaved into mature NGF by the protease MMP7 [115]. Accordingly, NGF might regulate follicular growth by promoting granulosa cell proliferation. Furthermore, several studies have suggested that NGF plays a modulatory role in the hormonal control of folliculogenesis. In vitro treatment with NGF induces the ovarian synthesis of FSH receptors, increasing the synthesis of androgens and estrogens [77] (Figure 5).

Alterations of the reproductive function, such as dysregulation in the synthesis and secretion of steroid hormones, are multifactorial and involve metabolic, endocrine, and anatomical factors. Since NGF contributes to normal follicular growth and a proper endocrine function, it is expected that altered NGF levels in the ovaries contribute to pathologies involving the reproductive and endocrine systems. Polycystic ovarian syndrome (PCOS) is one of the most common endocrine disorders in women of reproductive age and is among the most common causes of female infertility. PCOS is characterized by hyperandrogenism, chronic anovulation/oligoovulation, and polycystic ovarian morphology and is frequently accompanied by insulin resistance and MS [116].

Granulosa cells from PCOS patients secrete higher levels of NGF than healthy cells. The rise in NGF secretion results in a higher NGF concentrations in follicular fluid, which raise ovarian sensitivity to luteinizing hormone (LH)-like stimuli and promote androgen secretion [78]. Then, the overproduction of NGF can participate in the pathogenesis of PCOS. Concordant with this hypothesis, ovarian overexpression of NGF in a transgenic mouse model provokes the reproductive and metabolic features of PCOS [79]. Interestingly, testosterone production in response to gonadotropins was higher in ovaries obtained from transgenic mice with NGF overexpression, and ovaries from transgenic mice showed increased follicular recruitment and arrest. Paradoxically, while PCOS patients’ levels of NGF are high, serum levels of NGF in the same patients are decreased [117,118].

Of note, the hyperandrogenism observed in ovaries overexpressing NGF recapitulates the association between insulin resistance and hyperandrogenism in PCOS patients (up to 75% of PCOS patients have impaired insulin sensitivity). Whether insulin resistance is the cause or consequence of hyperandrogenism is still a matter of debate [119]. However, evidence supports hyperandrogenism as a significant driver of insulin resistance in PCOS patients. Hyperandrogenemia impaired the insulin sensitivity even without weight gain in a pharmacological model of PCOS [120].

The expression pattern of NGF and its receptors in male prenatal and adult rats and human testes suggests that NGF could influence the synthesis and secretion of steroid hormones by Leydig cells [80,81,82]. NGF expression is restricted to Leydig cells and, to a lesser extent, Sertoli cells, while NGF receptors are present predominantly on somatic cells in fetal testes [80,81]. Likewise, Leydig cells are the primary cell type in adult testes, expressing most of the NGF and its receptors, while germ cells show low expression levels of these genes [80,81,82].

In vitro treatment with NGF induces testosterone production by Leydig cells through the upregulation of key steroidogenic enzymes and stimulates cell proliferation by increasing cyclinD1 expression [121]. NGF administration induces dose-dependent testosterone secretion in azoospermic mice [122] while increasing the expression of the androgen-binding protein in rat testes [123].

On the other hand, human chorionic gonadotropin (hCG) induces TrkA expression in rat testes [124], while testosterone treatment represses NGF and NGFR expression [125,126]. Additionally, intramuscular NGF administration improves testosterone production in males with T2DM [126]. These findings suggest that NGF increases testosterone production and that there is a feedback loop between NGF and testosterone signaling. In contrast with the roles of NGF in MS development in females, the relevance of the NGF–testosterone loop in MS development in males remains to be explored.

In summary, NGF’s role within the gonad is to promote gametogenesis by acting on somatic cells. Additionally, NGF signaling modulates sex hormone secretion by the gonads. Both estrogen and testosterone regulate the fat distribution, energy expenditure, and food intake in both sexes [127]. Therefore, NGF’s actions toward the gonad influence reproduction and might indirectly regulate metabolic homeostasis in a sexually dimorphic manner. In females, high NGF production has been linked to hyperandrogenism, along with reproductive and metabolic dysregulation. Meanwhile, in males, low levels of NGF are linked to decreased testosterone levels and poor reproductive and metabolic outcomes.

## 7. Discussion

Metabolic syndrome increases the risk of CVD pathologies and T2DM. This is a multifactorial condition, and many mechanisms participate in developing the signs that define the syndrome. In this review, we collated the evidence about the effects of NGF signaling on the organs resulting from the immunoendocrine axis involved in developing MS. Moreover, we reviewed the evidence on the possible effects of dysregulated NGF levels described in MS patients (Table 1 and Table 2). Inconsistent NGF levels were noted in MS patients; some studies reported upregulated levels, while others found downregulation of this neurotrophin (Table 1).

These discrepancies might be due to the pleiotropic functions of NGF. Another explanation relates to the complexity of MS itself. Some epidemiological studies classify MS patients according to the duration or the number of signs present at the sampling time, while others do not make these distinctions. Some works describing elevated NGF levels considered MS patients with only three signs, corresponding to early MS. Others, meanwhile, considered long-term MS with more than four signs, showing low plasma NGF levels among such patients (Table 1). Finally, when the authors did not consider the progression of MS, the alterations in NGF levels were even more discrepant. More longitudinal studies, which offer greater value than transversal studies, will help to elucidate whether NGF dysregulation grows as MS progresses to a more severe condition.

The discrepancies in the mechanisms of NGF dysregulation during MS could be related to its multiple effects on metabolic organs. For instance, beta cells, adipose tissue, immune system cells, and the gonads (all part of the immunoendocrine axis involved in metabolic regulation) produce and secrete NGF. However, there are essential differences between these tissues in NGF production during overnutrition and T2DM. For instance, NGF production by beta cells in diabetic models recapitulates the effects on circulating NGF levels observed in MS patients. Higher levels of NGF are present shortly after SZT treatment in rats and during the early developmental age of db/db mice [87,88].

Moreover, the main actions of NGF in beta cells include islet morphogenesis and maturing, GSIS improvement, and apoptosis inhibition [49,50,53]. Thus, NGF could be an autocrine–paracrine protecting signal for beta cells under metabolic and cytotoxic stress conditions. However, because many parts of the body produce NGF, the actual contribution of pancreatic NGF to the circulating levels has yet to be discovered.

As previously mentioned, via at least two mechanisms, NGF increases GSIS in neonatal rat pancreatic beta cells. The first consists of the upregulation of genes involved in the coupling of glucose metabolism and insulin secretion, including Slc2a (GLUT2) Cacna1d, Cacna1c, Cacna1b, (voltage-dependent L-type calcium channel, alpha 1d, 1c and 1b subunits), and Cacna1a (voltage-dependent P/Q-type alpha 1A subunit) [128]. The second consists of the upregulation of Cacna1d [52] and the increased L-type calcium channel current density induced by NGF in neonatal beta cells [53,54].

In animal models with long-term MS, there are decreased levels of GLUT2 in the membrane and calcium currents [6,129]. Future works must assess how NGF secretion and signaling in pancreatic islets change during MS and T2DM development and during beta cell dysfunction. Since NGF signaling through TrkA increases GSIS, it could be of great interest to develop a synthetic TrkA agonist to restore GSIS in T2DM patients.

On the other hand, adipocytes are an active source of NGF, and it plays crucial functions in adipose tissue remodeling and metabolism [45,46,95]. Since adipose tissue greatly expands during obesity and MS, it is expected that this tissue contributes to a greater extent to increasing NGF levels during the early stages of MS than in later stages. NGF seems to have paradoxical effects on adipose tissue’s physiology, inducing angiogenesis, adipogenesis, and sympathetic innervation of WAT (factors associated with browning of adipose tissue and an oxidative phenotype of adipocytes).

In contrast, p75NTR induces adipose tissue fibrosis, blunts GLUT4 translocation and glucose uptake, and desensitizes catecholamine signaling (processes associated with obese, dysfunctional adipose tissue). As with most effects of NGF, the processes regulated by this factor in adipose tissue are highly context-dependent. More information is needed to understand whether NGF signaling through p75NTR is a driver of adipose tissue dysfunction and inflammation during obesity or if it is a consequence of adipocyte hypertrophy. Inducing MS in models of adipocyte-specific NGF deficiency will contribute to elucidating not only the relevance of adipocytes for circulating NGF levels but also the effects of disturbed NGF signaling during MS on other tissues such as beta cells, liver, and muscle.

The immune system is essential to metabolism control and tissue remodeling during MS. Immune cells reside in organs central to energy homeostasis, regulating their metabolic responses to environmental changes [75]. Notably, the immune system plays a pivotal role in adipose tissue homeostasis. Under normocaloric diets, immune cells in adipose tissue promote an anti-inflammatory state linked to a “healthy” metabolic profile and insulin sensitivity [66]. On the contrary, under hypercaloric conditions, adipose-tissue-residing immune cells develop an inflammatory phenotype, contributing to “unhealthy” changes in adipose tissue’s structure and function, which fuels the development of insulin resistance [5,10,67]. Immune cells are widely distributed NGF secretors, and they could play a significant role in affecting NGF signaling on tissues involved in developing MS.

Moreover, NGF secretion by several types of immune components depends on the context and molecular moieties in their microenvironment. This could contribute to the up- and downregulation of NGF depending on the stage of the disease. Accordingly, NGF deregulation in immune cells could explain the differences between the early and late stages of MS. Specifically, the presence of metabolic endotoxemia, which is present in some patients with MS, influences the inflammatory phenotype of tissue-residing immune cells [97]. Future research should address the cell-type-specific effects of NGF and how this neurotrophin favors communication between cells and tissues during MS development.

Sex hormones affect the development of the signs of MS, producing sexual dimorphism in the severity of these signs [130,131]. Thus, the effects of NGF signaling on the gonads and the synthesis of steroid hormones are important for fully understanding the effects of NGF on MS. The evidence demonstrates that dysregulation of NGF signaling in the ovaries participates in the development of PCOS and altered synthesis of sex hormones, a condition linked to insulin resistance and MS in women. Of note, in PCOS patients, the circulating NGF levels do not correlate with the intraovarian levels of this neurotrophin [116,117], supporting the notion that local NGF levels in specific tissues, rather than circulating levels, could help to elucidate the roles played by NGF during the development of alterations in metabolically important tissues. Moreover, the evidence in PCOS models suggests that increased NGF signaling in the ovaries could increase the risk of developing insulin resistance and MS in women. These ideas deserve further attention in clinical and animal models.

## 8. Conclusions

NGF signaling has been noted in some of the immunoendocrine tissues most relevant to MS development. Although there are several uncertainties about whether circulating NGF in MS patients is up- or downregulated, the mechanistic evidence points to the relevance of local, rather than systemic, NGF concentrations for the pathophysiology of MS, specifically by modulating the beta cell viability, GSIS, adipocyte metabolic profile, and inflammatory phenotype, and by influencing sex dimorphism in the development of MS signs. Future research is needed to fully understand the changes in NGF signaling at each stage of MS.

## Figures and Tables

**Figure 1 ijms-24-01957-f001:**
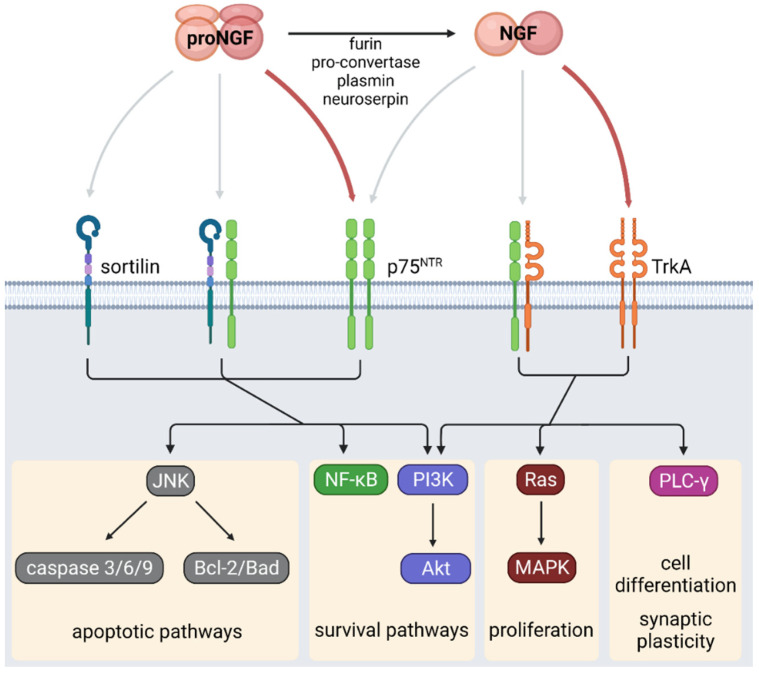
NGF signaling. NGF is synthesized as the precursor proNGF, which is cleaved by different proteases producing mature NGF. ProNGF binds with high affinity to the p75^NTR^ receptor while NGF preferentially binds to TrkA receptor. The local proNGF/NGF ratio determines the activation of apoptotic, cell survival, proliferation, or differentiation signaling pathways in different tissues. Created with BioRender.com.

**Figure 2 ijms-24-01957-f002:**
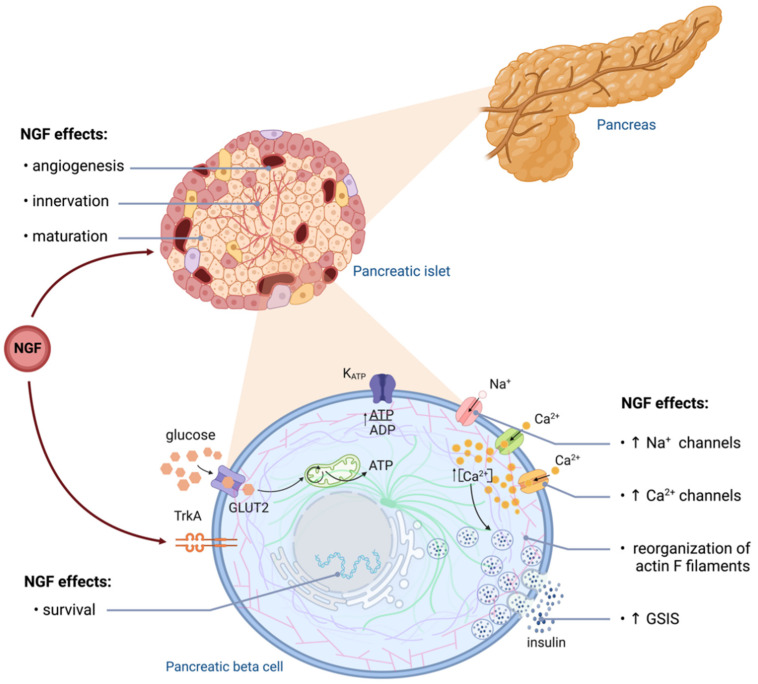
NGF effects on pancreatic beta cells. NGF participates in pancreatic islets’ maturation, favoring innervation and vascularization. In beta cells, NGF favors survival and upregulates key components of the insulin exocytosis process, which increases GSIS. NGF plays an essential role in the functional maturation of pancreatic beta cells from early stages of development. Created with BioRender.com.

**Figure 3 ijms-24-01957-f003:**
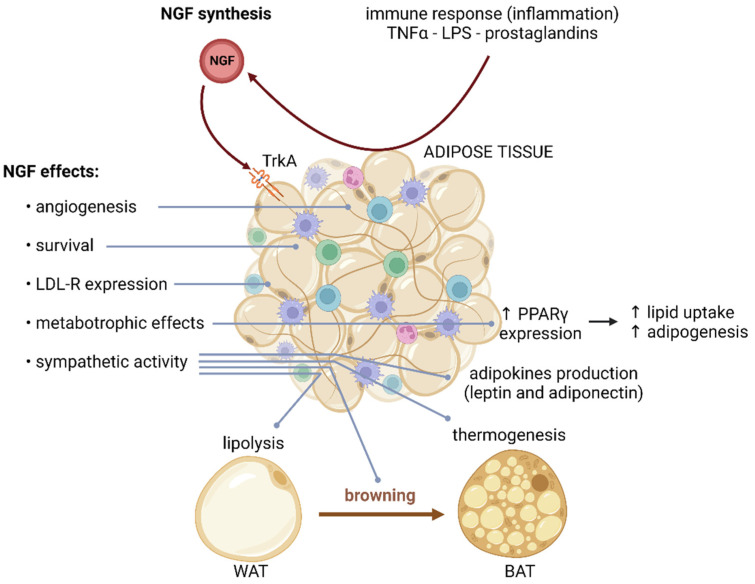
NGF effects on adipose tissue. Adipose tissue expresses NGF and NGF receptors. In adipocytes, proinflammatory factors induce an increase in NGF synthesis. TrkA signaling in adipose tissue favors angiogenesis and survival and has metabotropic effects. Through sympathetic innervation, NGF induces lipolysis in WAT, thermogenesis in BAT, and browning. Created with BioRender.com.

**Figure 4 ijms-24-01957-f004:**
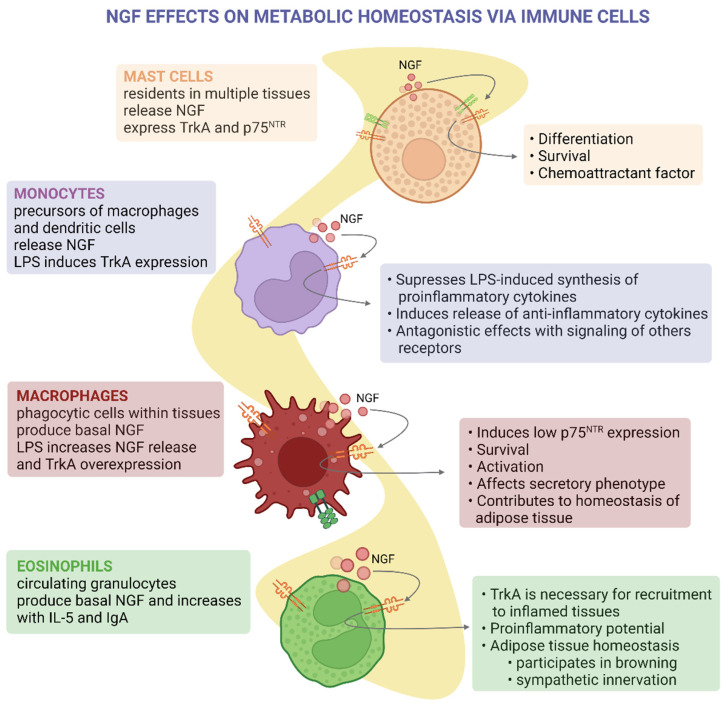
NGF effects on metabolic homeostasis via immune cells. NGF is secreted by different cells of the immune system. Under basal conditions or under different stimuli, these cells express NGF receptors. NGF, through the cells of the immune system residing in the tissues, in addition to regulating the immune response, also exerts actions on cell differentiation and survival and modulates the secretion of cytokines that regulate metabolism in the organs. Created with BioRender.com.

**Figure 5 ijms-24-01957-f005:**
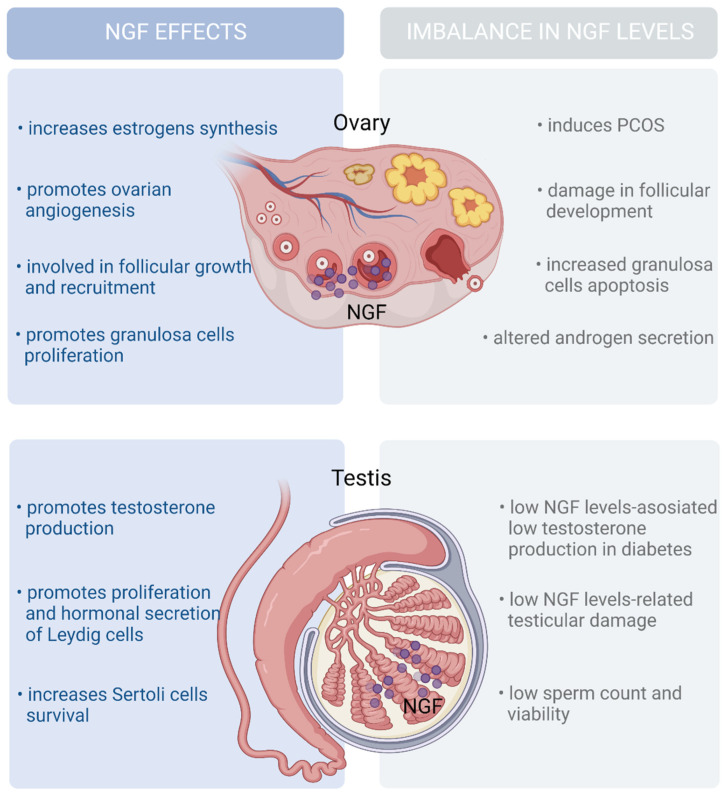
NGF effects on gonads. The ovaries and testes express NGF and its receptors. NGF participates in the development of reproductive function by modulating gametogenesis, angiogenesis, cell proliferation, and the secretion of steroid hormones in gonads. Alterations in NGF synthesis and signaling favor an imbalance in the synthesis of androgens and estrogens, which induces the development of PCOS, infertility, and tissue damage, among others. These abnormalities in sexual function have been linked to metabolic dysfunction such as insulin resistance or type 2 diabetes. Created with BioRender.com.

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
