# Peer review of "NGF and Its Role in Immunoendocrine Communication during Metabolic Syndrome"

_ijms, 2023, doi:10.3390/ijms24031957_

Round 1

Reviewer 1 Report (Previous Reviewer 2)

The revised manuscript still requires further improvement. Language and style needs extensive revision. For instance, repeats should be avoided, as 'NGF' in the abstract, 'promotes' at lines 459-464 and others, several sentences need to be reformulated (for instance lines 90, 107-8, 117), carefully revise all phrases in table 2 (for instance 'NGF expression and release NGF', 'NGF expression and release and it accumulates', same tense should be used throughout the table), lines 418, 421-2, low NGF levels-associate low testosterone (Figure 5), lines 471-3, 541, 543, 484 (extant?), 488-9, 563 etc. These are only few among many examples. I advise the authors to seek professional help to improve the language of the manuscript.

What is meant by adipose tissue matrix (line 362)?

The term 'resident' macrophages refers to the macrophages which are not deriving from recruited monocytes, the nature of which is just not starting to be understood. Hence, authors should use this term with caution.

The discussion is too long, authors should avoid repeating in the discussion what was previously mentioned. 

Author Response

Reviewer 1

The revised manuscript still requires further improvement. Language and style needs extensive revision.

R: We checked the grammar and style. The corresponding changes were made throughout the manuscript.

For instance, repeats should be avoided, as 'NGF' in the abstract,

R: We rephrased most of the abstract, which now reads as follows:

“Abstract: Nerve growth factor (NGF) was the first neurotrophin described. This neurotrophin contributes to organogenesis by promoting sensory innervation and angiogenesis in the endocrine and immune systems. Neuronal and non-neuronal cells produce and secrete NGF, and several cell types throughout the body express the high-affinity neurotrophin receptor TrkA and the low-affinity receptor p75NTR.  NGF is essential for glucose-stimulated insulin secretion and the complete development of pancreatic islets. Also, this factor is involved in regulating lipolysis and thermogenesis in adipose tissue. Immune cells produce and respond to NGF, modulating their inflammatory phenotype and the secretion of cytokines, contributing to insulin resistance and metabolic homeostasis. This neurotrophin regulates the synthesis of gonadal steroid hormones, which ultimately participate in the metabolic homeostasis of other tissues. Therefore, we propose that the imbalance in concentrations and signalling of this neurotrophin during metabolic syndrome contribute to its pathophysiology. In the present work, we aimed to describe the multiple roles of NGF in immunoendocrine organs that are important in metabolic homeostasis and related to the pathophysiology of metabolic syndrome.”

 'promotes' at lines 459-464 and others, several sentences need to be reformulated (for instance, lines 90, 107-8, 117),

R: We fixed typos throughout the manuscript.

carefully revise all phrases in table 2 (for instance, 'NGF expression and release NGF', 'NGF expression and release and it accumulates', the same tense should be used throughout the table),

R: We replaced the term release for secretion in table 2 to make this table clearer to the readers.

lines 418, 421-2, low NGF levels-associate low testosterone (Figure 5), lines 471-3, 541, 543, 484 (extant?), 488-9, 563 etc. These are only a few among many examples. I advise the authors to seek professional help to improve the manuscript's language.

R: We fixed the typos along the manuscript and homogenized the terms used throughout this work. The grammar and style were reviewed by a native speaker.

What is meant by adipose tissue matrix (line 362)?

R: We referred to the extracellular matrix of adipose tissue. This was fixed in the main text, and now the sentence reads as follows:

“NGF is secreted mainly by adipocytes and extracellular matrix of adipose tissue, with a minor contribution of vascular stromal-cells, including adipose tissue-residing macro-phages in biopsies of subcutaneous and omental adipose tissue from obese women.”

The term 'resident' macrophages refers to macrophages which are not derived from recruited monocytes, the nature of which is just not starting to be understood. Hence, authors should use this term with caution.

R: We agree with the reviewer about this point. Thus, we changed the term for macrophages settled in adipose tissue to state that these are macrophages that are found in this tissue. In line 41, the sentence reads as follows:

“The macrophages settled in adipose tissue promote the secretion of proinflammatory cytokines, inducing the development of a low-grade systemic inflammation, which also contributes to induce insulin resistance and other cardiometabolic alterations”

The discussion is too long; authors should avoid repeating in the discussion what was previously mentioned. 

R: We deleted the repeated information in the discussion section to make it clearer and shorter.

Reviewer 2 Report (Previous Reviewer 1)

In this review the authors report the available information regarding the impact of the NGF pathway on the immunoendocrine system in healthy and pathological conditions. Indeed the cell types and their associated NGF molecular mechanisms are usually not considered as the main ones when studying NGF, but are nevertheless critical for the metabolism, development and immune response. The authors clearly present the main functions of the NGF on the various cell types that they present and their associated pathologies.

Since the last submission the authors generated more figures and tables. As a consequence the key informations and potential discrepencies between reported publications are easily accessible. Finally, the authors have corrected the big weakness of their review being the discussion of the last version. Indeed the new version is more extensive and is well supported by the conclusion.   

Thus these efforts confirm my initial appreciation of this review as a valuable initiation to non-canonical effects of the NGF in the organism and in pathological cases. Only minor typos have to be corrected.

Author Response

Reviewer 2

In this review the authors report the available information regarding the impact of the NGF pathway on the immunoendocrine system in healthy and pathological conditions. Indeed the cell types and their associated NGF molecular mechanisms are usually not considered as the main ones when studying NGF, but are nevertheless critical for the metabolism, development and immune response. The authors clearly present the main functions of the NGF on the various cell types that they present and their associated pathologies.

Since the last submission the authors generated more figures and tables. As a consequence the key informations and potential discrepencies between reported publications are easily accessible. Finally, the authors have corrected the big weakness of their review being the discussion of the last version. Indeed the new version is more extensive and is well supported by the conclusion.   

Thus these efforts confirm my initial appreciation of this review as a valuable initiation to non-canonical effects of the NGF in the organism and in pathological cases. Only minor typos have to be corrected.

R: We appreciate your kind words about our manuscript. We fixed the typos and checked the grammar and style of the entire work.

This manuscript is a resubmission of an earlier submission. The following is a list of the peer review reports and author responses from that submission.

Round 1

Reviewer 1 Report

In this reviewe the authors report the available information regarding the impact of the NGF pathway on the immunoendocrine system in healthy and pathological conditions.  Indeed the cell types and their associated NGF molecular mecanisms  are usally not considered as the main ones when studying NGF, but are nevertheless critical for the metabolism, development and immune response. The authors clearly present the main fonctions of the NGF on the various cell types that they present and their associated pathologies. In this regard it is to emphisize that the manuscript is well written (except minors typos, see below), fluid and pleasant to read despite the amount of non-intuitive information reported. Thus this reviewing work provides clear, accessible and detailled pathways/pathology that are beyond the classical NGF scope and is with no doubt of great interest for the scientific community.  

All presented pathway and/or pathology are well referenced with both precursor and recent publications.  Moreover the number of non-canonical NGF expressing cell types presented in this manuscript, is pretty exhaustive.    

Despite the clear efforts of writting and clarity demonstrated in this manuscript some improvement can be made, especially for biologist naive to NGF:

1. Although the chosen cell types are diverse and pertinent, the authors should explain their choice in the introduction. Are they the only one known? the most studied/relevant in the immunoendocrine system?  Such a statement should improve both the focus and the interest of this review.

2. To my opinion the reproductive and immune system section deserve also dedicated figures, especially as they make reference to a various set of cell types.

3. It could be of great interest for the reader to have a summarizing table with all key information for each cell type in both healthy and pathological conditions. It could also give more weight to the discussion part that is certainly the weakness of this manuscript.

4. Indeed the Discussion is too short and do not confront the cell types and pathologies presented in the manuscript with the main ones that can be more intuitively associated with the NGF. This is a clear weakness for the the manuscript. Introducing stellate pancreatic cells and their association with cancer and chronic pancreatitis could be by example an interesting way to connect the presented pancreatic beta cell with more "common" NGF associated diseases. I would encourage the authors to clearly present in this section their reviewing work  as a counter weight to the extensive litterature concerning NGF and neurodegenerative disorder and cancer.   

Typos:

105 MGF maturation

172 suf-fering

196 p75NTR pritein

205 protein abun-dance

222 tolerance and at-tenuated

266 hormones and neu-rotrophins

296 NGF is both syn-the-

316 ex-pression of NGF receptors

317 ex-pressed under specific condition

333 stimulators of NGF such as F-κB

372 in a gon-adotropin free

Author Response

Reviewer 1

In this review the authors report the available information regarding the impact of the NGF pathway on the immunoendocrine system in healthy and pathological conditions.  Indeed the cell types and their associated NGF molecular mechanisms are usually not considered as the main ones when studying NGF, but are nevertheless critical for the metabolism, development and immune response. The authors clearly present the main functions of the NGF on the various cell types that they present and their associated pathologies. In this regard it is to emphasize that the manuscript is well written (except minors’ typos, see below), fluid and pleasant to read despite the amount of non-intuitive information reported. Thus, this reviewing work provides clear, accessible and detailed pathways/pathology that are beyond the classical NGF scope and is with no doubt of great interest for the scientific community.  

All presented pathway and/or pathology are well referenced with both precursor and recent publications.  Moreover, the number of non-canonical NGF expressing cell types presented in this manuscript, is pretty exhaustive.    

Despite the clear efforts of writing and clarity demonstrated in this manuscript some improvement can be made, especially for biologist naive to NGF:

  1. Although the chosen cell types are diverse and pertinent, the authors should explain their choice in the introduction. Are they the only one known? the most studied/relevant in the immunoendocrine system?  Such a statement should improve both the focus and the interest of this review.

Thank you for your comments. The review is re-focused on the role of NGF in regulating some cells of the immunoendocrine system with importance in metabolic homeostasis. Therefore, we review the regulation of NGF in adipose tissue and pancreatic beta cells due to their relevance in the synthesis of adipokines and insulin, respectively. We will also review the roles of NGF as a modulator of the release of pro-and anti-inflammatory molecules by some immune system cells. Finally, we will describe the regulation of NGF in the synthesis and secretion of sex steroid hormones. The focus will be on how these pro- and anti-inflammatory molecules and sex steroid hormones can interfere with insulin signaling and thus with metabolic homeostasis.

  1. To my opinion the reproductive and immune system section deserve also dedicated figures, especially as they make reference to a various set of cell types.

The review focuses on the role of NGF in regulating some cells of the immunoendocrine system with importance in metabolic homeostasis. Because it was too long, we will only describe mechanisms and cell types related to the synthesis and secretion of pro- and anti-inflammatory molecules and sex steroid hormones.

  1. It could be of great interest for the reader to have a summarizing table with all key information for each cell type in both healthy and pathological conditions. It could also give more weight to the discussion part which is certainly the weakness of this manuscript.

We included three figures that compile the most relevant information for each cell type.

  1. Indeed the Discussion is too short and do not confront the cell types and pathologies presented in the manuscript with the main ones that can be more intuitively associated with the NGF. This is a clear weakness for the manuscript.

You are right; the discussion was not good at all. We discussed the information in each section and made a brief conclusion. In the end, we mention some relevant aspects for a better understanding of the role of NGF in regulating metabolic homeostasis and the pathologies related to poor regulation by NGF.

Introducing stellate pancreatic cells and their association with cancer and chronic pancreatitis could be by example an interesting way to connect the presented pancreatic beta cell with more "common" NGF associated diseases. I would encourage the authors to clearly present in this section their reviewing work as a counterweight to the extensive literature concerning NGF and neurodegenerative disorder and cancer. 

We appreciate the proposal, however, due to the approach, we decided not to mention these pathologies because they are not directly related to the loss of systemic metabolic homeostasis. 

We make corrections to typographical errors, like the following and others, thank you!

105 MGF maturation

172 suf-fering

196 p75NTR pritein

205 protein abun-dance

222 tolerance and at-tenuated

266 hormones and neu-rotrophins

296 NGF is both syn-the-

316 ex-pression of NGF receptors

317 ex-pressed under specific condition

333 stimulators of NGF such as F-κB

372 in a gon-adotropin free

Reviewer 2 Report

The present manuscript reviews the role of the Nerve Growth Factor (NGF) in the endocrine and immune system, with a focus on its role in the pancreas, the adipose tissue, the reproductive system and immune cells. The manuscript is broad, unfocused and rather superficial, without analyzing in depth many of the discussed topics. The authors should discuss in greater detail the literature, the implicated mechanisms and draw clearer conclusions about the role of NGF, proNGF or their receptors in the different systems.

Most citations are wrong, it seems that they are shifted, the last reference (160) is not cited.

Original articles instead of reviews should be cited to support information on the topic discussed.

Extensive language editing is required. Language style needs significant improvement. Syntax should be corrected and rephrasing requiredat many places.

Paragraphs need to be reorganized.

Repeats should be avoided.

The introduction has many paragraphs consisting each of 1-2 sentences. The first two paragraphs of the introduction are too general and not necessarily related to the topic of the review.

Figures: instead of Figure 1 which has been included in many other reviews, another two figures related to the role of NGF in the reproductive system and the immune system could be included.

The discussion is too short, it should draw conclusions on the functions of NGF, discuss future perspectives, therapeutic aspects, as well as limitations.

The role of NGF in mast cells should be discussed.

Regulation of neutrophil and eosinophil recruitment by TRKA (PMID: 31980574, PMID: 31871023) should be cited.

The role of NGF/TrkA signaling in microglia function should be mentioned: PMID: 28894299, PMID: 30817930, PMID: 35327616, PMID: 29473218

P75 expression in adipose tissue progenitor subsets (CD45-/CD34+/CD31-) should be mentioned (PMID: 31186409). Also the interaction of tumor cells with adipocyte progenitors through NGF could be discussed (PMID: 34408135).

It is Nerve (not Neuronal) Growth Factor

Author Response

Reviewer 2

The present manuscript reviews the role of the Nerve Growth Factor (NGF) in the endocrine and immune system, with a focus on its role in the pancreas, the adipose tissue, the reproductive system and immune cells. The manuscript is broad, unfocused and rather superficial, without analyzing in depth many of the discussed topics. The authors should discuss in greater detail the literature, the implicated mechanisms and draw clearer conclusions about the role of NGF, proNGF or their receptors in the different systems.

Thanks for the comments. We delimited the review on the role of NGF in regulating some cells of the immunoendocrine system with importance in metabolic homeostasis.

We include information from articles on NGF, proNGF, and their receptors that focus on adipose tissue and the pancreatic beta-cell, due to their relevance in the synthesis of adipokines and insulin. In addition to the synthesis and release of pro-and anti-inflammatory molecules by some immune system cells. Finally, we describe the regulation of NGF in the synthesis and secretion of sex steroid hormones.

- Most citations are wrong, it seems that they are shifted, and the last reference (160) is not cited.

We appreciate the observation. The sequence of the references has been corrected.

- Original articles instead of reviews should be cited to support information on the topic discussed.

More original references were included.

Extensive language editing is required. Language style needs significant improvement. Syntax should be corrected and rephrasing require dat many places.

Paragraphs need to be reorganized.

Repeats should be avoided.

We reorganized paragraphs and sections and made it easier, trying not to be repetitive.

The introduction has many paragraphs consisting each of 1-2 sentences. The first two paragraphs of the introduction are too general and not necessarily related to the topic of the review.

We changed the introduction, focusing on the role of NGF in regulating some cells that are part of the immunoendocrine system with importance in metabolic homeostasis.

Figures: instead of Figure 1 which has been included in many other reviews, another two figures related to the role of NGF in the reproductive system and the immune system could be included.

We included two more figures. The first figure integrates NGF information in the regulation of some cells of the immune system. The second figure integrates the effects of NGF on gonads. The information in figure two was separated. We have now one figure on the effects of NGF on pancreatic beta cells and another on the effects of NGF on adipose tissue.

The discussion is too short, it should draw conclusions on the functions of NGF, and discuss future perspectives, therapeutic aspects, as well as limitations.

You are right; the discussion was not good at all. We discussed the information in each section and made a brief conclusion. In the end, we mention some relevant aspects for a better understanding of the role of NGF in regulating metabolic homeostasis and the pathologies related to poor regulation by NGF.

The role of NGF in mast cells should be discussed. Regulation of neutrophil and eosinophil recruitment by TRKA (PMID: 31980574, PMID: 31871023) should be cited.bv The role of NGF/TrkA signaling in microglia function should be mentioned: PMID: 28894299, PMID: 30817930, PMID: 35327616, PMID: 29473218

P75 expression in adipose tissue progenitor subsets (CD45-/CD34+/CD31-) should be mentioned (PMID: 31186409). Also, the interaction of tumor cells with adipocyte progenitors through NGF could be discussed (PMID: 34408135).

We included and discussed all your suggestions, thank you. We focused on the regulation of metabolic homeostasis outside the central nervous system.

It is Nerve (not Neuronal) Growth Factor

Thank you for the correction.

Reviewer 3 Report

This review is basically narrative and adds almost none of new comprehensive knowledge on NGF and Trk system in non-neuronal tissues. As already everybody knows that NGF-Trk signaling pathway in neuronal system plays an important role for neuronal differentiation and survival of neuronal cells, this signaling pathway in non-neuronal cells might act as a cell-proliferation mediator. What is the molecular basis of these big difference in the signal transduction pathways? Unfortunately, the authors did not discuss about possible mechanisms for these differences, but only described the respective situation of different non-neuronal tissues. In other wards, there was no mechanistic discussion in the present manuscript. This reviewer also found many gramatical and typographical errors which makes me extremely difficult to understand and follow their logic. Unfortunately, this reviewer is negative against this review article. 

Author Response

Reviewer 3

This review is basically narrative and adds almost none of the new comprehensive knowledge on NGF and Trk system in non-neuronal tissues. As already everybody knows that the NGF-Trk signaling pathway in the neuronal system plays an important role in neuronal differentiation and survival of neuronal cells, this signaling pathway in non-neuronal cells might act as a cell-proliferation mediator. What is the molecular basis of these big differences in the signal transduction pathways? Unfortunately, the authors did not discuss possible mechanisms for these differences, but only described the respective situation of different non-neuronal tissues. In other words, there was no mechanistic discussion in the present manuscript. This reviewer also found many grammatical and typographical errors which makes me extremely difficult to understand and follow their logic. Unfortunately, this reviewer is negative about this review article. 

We welcome feedback, thanks for your comments. We tried to look for another review of the kind of what we wrote and could not find it. We tried to improve it, and we hope you will like the new approach.

We reviewed and clarified the focus was on the role of NGF in regulating some cells that are part of the immunoendocrine system with importance in metabolic homeostasis. Therefore, we review works on NGF regulations in adipose tissue and pancreatic beta cells due to their relevance in the synthesis of adipokines and insulin. We will also review the role of NGF as a modulator of the release of pro-and anti-inflammatory molecules by some immune system cells. Finally, we describe the regulation of NGF in the synthesis and secretion of sex steroid hormones. The focus will be on the context of how these molecules, pro-and anti-inflammatory, and sex steroid hormones, can interfere with insulin and adipokine signaling as well as metabolic homeostasis.

We discussed each section of the information and made a brief conclusion. We included some relevant aspects for a better understanding of the role of NGF in regulating metabolic homeostasis and the pathologies related to poor regulation by NGF. Some paragraphs and sections were reorganized, in addition to correcting typographical errors.

Round 2

Reviewer 2 Report

The manuscript has very little improved through revisions. 

It is obvious that different parts of it were written by different authors: while the 'NGF in beta cells' is ok, the parts on adipose tissue and immune cells are not publishable. 

The text contains numerous mistakes, wrong information, linguistic mistakes etc 

The manuscript in its whole should be carefully revised by the senior author and professional language check is required. 

Figures 3 and 4 present wrong information and should be carefully revised, every phrase/statement should be backed-up by literature...

Introductions on what is the immune system, innate immunity are obviously unnecessary and only lower the quality of the manuscript. 

The connection the authors try to establish between NGF and immune cells in the adipose tissue is only based on assumptions and cannot be part of a review. For instance, to my knowledge, the role of NGF in adipose tissue macrophages or the connection between NGF, eosinophils and beiging/browning has not been shown. The authors should be careful with what they state. 

Still, the manuscript remains unfocused and superficial. One topic, for instance NGF in beta cells or NGF in immune cells could be chosen and analyzed in greater depth.  

Reviewer 3 Report

The revised manuscript looks still narrative and basically adds none of new way of thinking. This reviewer unfortunately could not find any merit of the publication of this review article.